# Transmembrane Protein 43: Molecular and Pathogenetic Implications in Arrhythmogenic Cardiomyopathy and Various Other Diseases

**DOI:** 10.3390/ijms26146856

**Published:** 2025-07-17

**Authors:** Buyan-Ochir Orgil, Mekaea S. Spaulding, Harrison P. Smith, Zainab Baba, Neely R. Alberson, Enkhzul Batsaikhan, Jeffrey A. Towbin, Enkhsaikhan Purevjav

**Affiliations:** 1The Heart Institute, Department of Pediatrics, University of Tennessee Health Science Center, Memphis, TN 38103, USA; borgil@uthsc.edu (B.-O.O.); neely.alberson@lebonheur.org (N.R.A.); jtowbin1@uthsc.edu (J.A.T.); 2Children’s Foundation Research Institute, Le Bonheur Children’s Hospital, Memphis, TN 38105, USA; 3College of Medicine, University of Tennessee Health Science Center, Memphis, TN 38163, USA; mmill268@uthsc.edu (M.S.S.); hsmit143@uthsc.edu (H.P.S.); 4Rhodes College, Memphis, TN 38112, USA; zbaba1@uthsc.edu; 5Department of Pharmacology, Addiction Science, and Toxicology, University of Tennessee Health Science Center, Memphis, TN 38163, USA; ebatsaik@uthsc.edu; 6Pediatric Cardiology, St. Jude Children’s Research Hospital, Memphis, TN 38105, USA

**Keywords:** arrhythmogenic cardiomyopathy, auditory neuropathy spectrum disorders, cancer, *LUMA*, skeletal myopathy, *TMEM43*, transmembrane protein 43

## Abstract

Transmembrane protein 43 (*TMEM43* or *LUMA*) encodes a highly conserved protein found in the nuclear and endoplasmic reticulum membranes of many cell types and the intercalated discs and adherens junctions of cardiac myocytes. TMEM43 is involved in facilitating intra/extracellular signal transduction to the nucleus via the linker of the nucleoskeleton and cytoskeleton complex. Genetic mutations may result in reduced *TMEM43* expression and altered TMEM43 protein cellular localization, resulting in impaired cell polarization, intracellular force transmission, and cell–cell connections. The p.S358L mutation causes arrhythmogenic right ventricular cardiomyopathy type-5 and is associated with increased absorption of lipids, fatty acids, and cholesterol in the mouse small intestine, which may promote fibro-fatty replacement of cardiac myocytes. Mutations (p.E85K and p.I91V) have been identified in patients with Emery–Dreifuss Muscular Dystrophy-related myopathies. Other mutations also lead to auditory neuropathy spectrum disorder-associated hearing loss and have a negative association with cancer progression and tumor cell survival. This review explores the pathogenesis of *TMEM43* mutation-associated diseases in humans, highlighting animal and in vitro studies that describe the molecular details of disease processes and clinical, histologic, and molecular manifestations. Additionally, we discuss *TMEM43* expression-related conditions and how each disease may progress to severe and life-threatening states.

## 1. Introduction

Transmembrane protein 43 (*TMEM43*, NM_024334), first described as *LUMA*, is a gene that encodes a highly conserved integral protein of 45 kDa, expressed in the nuclear envelope and in endoplasmic reticulum (ER), which plays critical roles in maintaining inner nuclear membrane (INM) stability of many organs [1,2,3]. More recently, *TMEM43*’s expression was also found in composite and adherens junctions and intercalated disks of cardiac myocytes and epithelial cells in humans and other mammals [4]. Additionally, *TMEM43* has been identified in cochlear glia-like supporting cells of the two-pore-domain potassium (K2P) channels that are permeable to sodium (Na^+^), potassium (K^+^), and cesium (Cs^+^) ions, suggesting a role for *TMEM43* in the functioning of ion channels [5]. Genetic mutations and gene expression variations in *TMEM43* have been associated with many human diseases, including arrhythmogenic cardiomyopathy (ACM), formerly known as arrhythmogenic right ventricular dysplasia/cardiomyopathy (ARVD/C) [6,7], Emery–Dreifuss Muscular Dystrophy (EDMD) [8,9], and auditory neuropathy spectrum disorders (ANSDs) [10,11], as well as the progression of various cancers in humans [12,13,14]. A haploinsufficiency of *Tmem43* in murine cardiac myocytes has been shown to cause senescence-associated cardiomyopathy via the DNA damage response (DDR) pathway [15]. While downregulation of *Tmem43* has been shown to play a critical role in cardiac hypertrophy via epithelial-to-mesenchymal transition in mouse models [16], TMEM43’s upregulation protects against lipopolysaccharide (LPS)-induced cardiac injury via inhibiting ferroptosis [17]. Gene-silencing of *TMEM43* in the cochlea has been associated with a loss of K^+^ conductance currents in cochlear supporting glia-like cells (GLSs). Moreover, TMEM43 was identified to activate the nuclear factor kappa B (NF-kB) pathway through epidermal growth factor receptor (EGFR) signaling or stabilize pre-MRNA processing factor 3 (PRPF3), thus promoting cancer cell proliferation, survival, and migration [12,14].

All these reports point out the important molecular and functional properties of *TMEM43*. In this review, we collected key findings of *TMEM43* research in humans and animal and cellular models. Our investigation demonstrates that genetic mutations and gene expression levels of *TMEM43* play important roles in the development and pathophysiology of various diseases. Our review emphasizes the critical rationale for promoting future research on assessing *TMEM43* expression for preventive and diagnostic tests and personalized treatment strategies in *TMEM43*-associated diseases.

## 2. Biological and Molecular Functions of TMEM43

TMEM43 is an integral membrane protein that has primarily found in the INM and ER membranes in many cell types [1]. It is highly conversed across various species, including unicellular eukaryotes, insects, plants, bacteria and vertebrates. In humans, TMEM43 mRNA is differentially expressed in tissues with the highest levels found in the placenta and lower levels found in the small intestine, spleen, prostate, and testes [4]. Expression of TMEM43 is also found to be part of the cytoplasmic plaques in the zonula adherens and punctum adherens of various mammalian epithelial cells, as well as the plaques of composite junctions in intercalated discs of cardiac myocytes (Figure 1) [18]. In addition, TMEM43 has recently been found in ion channels in GLSs residing in the cochlea of the inner ear [5].

The TMEM43 protein belongs to the family of tetraspanin proteins that contain four transmembrane (TM1-4) domains (Figure 1), span through nuclear, ER, or cellular membrane, and carry out critical biological functions essential for cell migration, membrane integrity, protein trafficking, and signal transduction [19]. Four TM1-4 spanning domains are thought to play a role in self-oligomerization, while the TM3 and Loop2 domains contain the pore-forming residue of TMEM43 [1,10]. The N- and C-termini of TMEM43 are localized in nuclear and cytoplasmic portions of the cell, while a large hydrophilic domain resides mostly in the perinuclear space (PNS) [1]. The portion of the TMEM43 protein localized in the cytoplasm or ER lumen contains about 250 amino acid residues [20]. In addition, the specific residues have been predicted as the attachment sites of interacting protein that are involved in TMEM43’s post-translational modifications, such as protein phosphorylation (Ying-Yang motif sites), N-linked glycosylation at the 356 residues of the O-glycosylation site [21], and SUMOylation at the 266–269 residues of the small ubiquitin-like modifier (SUMO) site using the online tools via the ExPASy website [6]. Conservation of all those residues relevant for the proper function and structure of TMEM43 signifies its essential nonredundant role in cellular functions. A monomeric structure of TMEM43 is available through network-based AlphaFold2 machine-learning modeling [22]; however, its intramembrane domain contains an incomprehensible scrambled residue. Therefore, determination of the detailed architecture of the TMEM43 protein and structural correlations using cryo-electron microscopy or crystal structure analysis is critical to understand its pleiotropic functions.

The nuclear envelope is composed of an outer nuclear membrane (ONM) and an INM with the PNS lying in between, and the ER membrane being contiguous with the ONM (Figure 2) [23]. The two membranes are frequently fused at the site of nuclear pore complexes, which allow for bidirectional trafficking of molecules between the cytoplasm and nucleus [24]. The INM functions as a critical regulator of genomic integrity and is lined by nuclear lamina, which are polymers of intermediate filament-type lamin proteins associated with a number of integral membrane proteins that work together to maintain nuclear compartmentalization and integrity [25]. The cellular nucleus is linked to the cytoplasm via the linker of nucleoskeleton and cytoskeleton complex (LINC), which is made of proteins containing Klarsicht, ANC1, Syne Homology (KASH) domains and Sad1 and UNC-84 (SUN) domains [26,27]. In mammals, nuclear envelope nesprins, a large family of spectrin repeat transmembrane proteins encoded by the *SYNE1* and *SYNE2* genes, represent the critical molecules of LINC connecting the ONM to the actin cytoskeleton via their KASH domains. The LINC also interacts with SUN1 and SUN2 proteins, linking nesprins to the INM [28]. Underlying the INM, lamins contribute to anchoring LINC components to the nuclear envelope, granting a key role for the LINC in the force-transmitting at the nuclear envelope, as well as in regulating the major cellular processes [29]. LINC connections between the nucleus and cytoskeleton are essential for a wide range of cellular functions and play a role in mechanosensation by transmitting intra and extracellular mechanical stimuli via the cytoskeleton to the nucleus [29,30]. More specifically, mechanical cues—such as physical forces or alterations in the extracellular membrane—are translated into biochemical signals [31]. Changes in biochemical signals may influence gene expression through chromatin location and organization. The TMEM43 protein is shown to interact with lamins A/C and B1, emerin, and SUN2 at the INM [1]. Binding to type-A lamins is responsible for its localization to the INM, while TMEM43 is thought to play a critical role in emerin localization. Emerins are ubiquitously expressed in tissues and predominately localized to the nuclear envelop in skeletal and cardiac muscle; they mediate the anchorage of INM to the cytoskeleton [32], transduce mechanosensitive signals to strain stimuli, and regulate gene expression responses through β-catenin interactions to the intercalated disks located at the cell–cell connections [33].

More recent reports have demonstrated the localization of TMEM43 protein at the intercalated disc of cardiac myocytes [4], while Christensen et al. found TMEM43 mainly localized at the sarcolemma [7]. Sarcolemma is the membrane of muscle cells. In contrast to skeletal muscles, cardiomyocytes are connected to adjacent cardiomyocytes by the intercalated disc at each end. The intercalated disc is a specialized interdigitating cell membrane containing three distinguished structures: (1) adherens junctions comprised by N-cadherin, catenins and vinculin (VCL); (2) desmosomes comprised by desmoplakin (DSP), desmocollin (DSC), desmoglein (DSG), junctional plakoglobin (JUP), plakophilins (PKP) and desmin intermediate filament; and (3) gap junctions containing connexins (CX) (Figure 3). Hence, mutations in proteins of the desmosomes, adherence and gap junctions could lead to a “domino effect” disruption of the intercalated disc integrity and cell–cell connections, creating arrhythmogenic substrate in the heart. Supporting this, the *TMEM43* p.S358L missense variant—a causal mutation for arrhythmogenic right ventricular cardiomyopathy type-5 (ARVC5)—has been shown to disturb a conduction velocity at the gap and adherens junctions due to zonula occludens-1 (ZO-1) protein ablation from cardiac myocyte cell–cell junctions [34]. This results in redistribution of JUP and α-catenin to the cytoplasm from cell–cell junctions and alteration in CX43 phosphorylation at the gap junctions [35].

Recently, studies by Klinke et al. determined that the hydroxyl groups at the Drosophila CG8111 serine-333 residue homologous to human S358 TMEM43 residue plays a critical role in maintaining proper protein structure and function by forming hydrogen bonds between TM3 and TM4 helices (Figure 4) [20]. Moreover, the majority of mutant TMEM43 S358L molecules have been shown to accumulate in the nuclear membrane of cardiomyocytes in an unglycosylated state, leading to altered gene expression [21]. An association of ARVC5 and TMEM43 with peroxisome proliferator response elements (PPREs) has also been predicted [36], but the mechanisms by which this mutation alters the functions of PPREs and its interacting proteins in ARVC5 remains unclear. Thus, further molecular studies clarifying TMEM43’s effects on its interacting protein’s nuclear position and function, gene expression, intracellular force transmission, and cell–cell connections will shed light on understanding the mechanisms of ARVC5 that develop as result of the S358L missense substitution.

Intriguing data shows an abundance of the *TMEM43* mRNA expression in human placenta, suggesting a significant role in placental development and function. The ELGAN cohort study of 386 placentas collected from extremely preterm infants identified a significant increase in expression levels of 48 placental mRNAs, including *TMEM43*, that were associated with late cognitive impairment diagnosed at age 10 in these children [38]. This study defined a potential link between *TMEM43* expression and placental inflammatory pathways and suggested neurodevelopmental risks in preterm infants through *TMEM43*-mediated NF-κB activation.

In summary, TMEM43 protein’s highly conserved structure and various localizations are vitally important for its diverse functions. The tetraspanin architecture enables execution of various critical roles for cellular homeostasis as a structural sarcolemmal and nuclear envelope protein, an ion channel and gap junction complexes organizer, and as a mediator of signal transduction pathways. The deleterious consequences of genetic and posttranslational abnormalities of TMEM43 underscore indispensable missions of its structure-function relationships in the pathogenesis of a spectrum of human health and disease states affecting multiple organ systems and ranging from severe ARVC5 to muscular dystrophies, hearing loss, and various cancers.

## 3. Arrhythmogenic Cardiomyopathies

Arrhythmogenic cardiomyopathies (ACMs) is a heterogeneous set of diseases that have serious cardiac pathologies and outcomes, including life-threatening arrhythmias, sudden cardiac death (SCD), and heart failure (HF) [39,40]. Various studies in diverse populations and geographical locations using different approaches estimated the population frequency of ACM to be from 1:1000 to 1:5000 [41,42]. While first reports of diffuse fibrotic remodeling and failure of the right ventricle (RV) date back to as early as 1961 [43], the first systematic report of the disease has garnered interest as an arrhythmogenic right ventricular dysplasia (ARVD), which later has been named as an arrhythmogenic right ventricular cardiomyopathy (ARVC) [44,45,46]. A classical feature of ACMs is the presence of symptomatic arrhythmia and diagnosis generally occurs after the individual presents with arrhythmia [47,48,49], particularly with SCD in young people [50]. Patients commonly present with this disease between the 2nd to 4th decades of life, which can have initial unfavorable manifestations and outcomes with SCD [40,51,52]. Individuals who participate in organized athletics or frequently train with high endurance activities can be at higher risk for symptomatic presentation of the disease; research is frequently performed on this population [53,54,55]. Children can also present with ARVC disease, usually during the late second decade, and are generally at a higher risk of SCD from a possible correlation with higher physical activity level [56]. Significant sex differences in disease onset and penetrance, clinical severity, arrhythmic risks and outcomes have broadly been documented [57,58]. Elevated levels of testosterone compared to that in healthy controls have been found in male patients with ACM and plasma testosterone levels higher than 13.5 nmol/L predicted significant risks for adverse outcomes in male patients. In female patients, lower estradiol levels were correlated with the major adverse arrhythmic events. Protective mechanisms of estrogen include anti-apoptotic effects, reduction in oxidative stress, lipogenesis and lipid accumulation in the heart. On the other hand, testosterone has been shown to promote ACM severity through increased cardiomyocyte apoptosis and lipid accumulation [58].

When diagnostically evaluating ARVC, four stages are generally described in the literature [59,60,61]. In the early stages of the disease, the only symptom of this devastating disease is life-threatening arrhythmias, which typically manifest during adolescence with an initial event of ventricular tachyarrhythmias (VTAs) [53]. In the earliest phase, no changes suggestive for ARVC are typically observed in the electrical, structural, or histological tests, including cardiac magnetic resonance (CMR) imaging. Since individuals can experience sustained VTAs, there is a risk of SCD in the absence of ventricular dysfunction during the first stage of the disease, highlighting the need for mechanistic understanding to screen “at risk” individuals earlier [59]. The second stage is marked by symptomatic arrhythmias characteristic of the disease, and this stage is often followed by the RV and biventricular failure as the third and fourth stages, respectively. This may potentially be due to a sequalae of events involving electric abnormalities that precede myocyte death and adipogenic changes with fibro-fatty infiltrations in the myocardium [62].

Diagnostic criteria within this clinically heterogeneous and complex disease have evolved as the diagnostic imaging modalities and genotyping strategies have progressed [63,64]. In addition to classical ARVC with right-sided pathological phenotypes, left ventricular (LV)-sided or ALVC, and biventricular pathologies have been recognized [65,66,67]. Coexistence of ALVC and biventricular disease patterns has been observed in 67% of families [46]. The 2010 update of the 1994 International Task Force Criteria has been developed to establish a clinical diagnosis of ARVC [51], which included detailed diagnosis of the right-sided ARVC based on CMR imaging; however, arrhythmogenic left ventricular cardiomyopathy (ALVC) or bi-ventricular ACMs are not included. Much progress has been made in the 2019 Heart Rhythm Society (HRS) expert consensus statement [49] along with the development of new criteria called the Padua Criteria [68,69] and the European Task Force consensus [70], representing complementary yet distinct and interconnected clinical approaches for addressing risk stratification, diagnostics, and management of ACM. These new diagnostic criteria have successfully addressed critical limitations in previous 2010 diagnostic Task Force approaches by including the full spectrum of advanced imaging and diagnostic tools for LV and biventricular forms of ACM phenotypes, in addition to classic ARVC. As evidenced by the inability to easily detect the disease in its early stages, more considerations should be made in advanced CMR imaging and genetic profiling [49,71,72].

Genetically, the disease is related to various mutations in desmosomal genes and other components of the intercalated discs in cardiomyocytes [61,73,74,75,76,77,78,79]. Non-desmosomal genetic bases of ACMs include mutations identified in genes encoding cytoskeletal [80,81], transmembrane and nuclear proteins including TMEM43 [6], intermediate filament, transforming growth factor beta 3 (TGF-β3), and ion channel proteins [18,82,83] that interact with desmosomal proteins, suggesting that genetically induced disruption in integrity of the intercalated discs is one of the key factors in the “final common pathway” that fosters the development of ACM [74]. The inheritance pattern is considered to have autosomal dominant transmission and studies have shown variability in the penetrance (e.g., with some cases of incomplete penetrance) and expressivity of the disease based on the affected genes [74,84,85,86]. Several studies demonstrated cases of ACM inherited in an autosomal recessive fashion, with some also involving the skin and/or isolated LV involvement [87].

Fibro-fatty replacement of the ventricular myocardium is a histological key of the disease, which has been considered as a morphological substrate that leads to the cycle of fatal arrhythmias and SCD [88]. The RV apex and outflow tract are typically affected with right-sided pathology, while left-sided pathology involves the subepicardial and mid-wall LV with mild to severe dilation. Biventricular ACM displays fatty infiltration of both ventricles including the interventricular septum (IVS) and biventricular dilation (Figure 5) [89]. The life-threatening VTAs or SCD may precede structural remodeling with fibro-fatty infiltrations of the myocardium or clinical evidence of HF [90]. Later, fibrofatty infiltrations promote electric instability and impair ventricular mechanical function, creating the feedback loop for exacerbated arrhythmias and progressive HF [89,91,92,93].

Varied theories explaining the pathogenesis of ACM exist. Several studies have specifically analyzed the pathogenesis of ARVC with enhanced adipogenesis and fibro-fatty remodeling through desmosome-associated alterations in Wnt/β-catenin signaling, peroxisome proliferator-activated receptor (PPAR) signaling, Hippo signaling, defects in ion channels, and DDR pathways [15,62,92,94,95,96]. Desmosomal proteins (DSP, JUP, PKP2, DSG and DSC) cooperatively assemble with other proteins at cell–cell junctions that are responsible for coupling of myocytes in the heart [97]. This connection provides continuous cell-to-cell links between cardiac myocytes and to the sarcomeric actin and intermediate filament desmin (Figure 3) [74,98,99]. Beta (β)-catenin is an effector in canonical Wnt signaling pathway with structural and functional similarly to JUP, also known as gamma (γ)-catenin [94]. In a murine study, Garcia-Gras et al. mutated the desmosomal DSP protein, theorizing that protein assembly would be inhibited, which would thus free γ-catenin. By doing so, γ-catenin was free to translocate to the nucleus and competitively inhibit β-catenin, suppressing canonical Wnt signaling and thereby promoting cardiac myocyte adipogenesis, fibrogenesis, and apoptosis that resulted in ARVC in vivo. A separate colony of DSP-deficient mice via deletion of the *Dsp* gene had myocardial destruction and ventricular tachycardia in addition to cardiac myocyte adipogenesis, fibrogenesis, and apoptosis. Deletion of *Dsp* also confirmed that JUP protein localized to the nucleus in *Dsp*^−/−^ mice compared to cytoplasm localization in *Dsp*^+/+^ mice. The resulting adipogenesis and fibrogenesis in these studies was due to a transcription switch that leads to upregulation of PPARγ and CEBPα (CCAAT enhancer-binding protein alpha) transcriptional factors and their target genes adiponectin and lipoprotein lipase, which was confirmed by RT-PCR and immunofluorescence analysis [94]. Additionally, perturbed AKT1, Wnt/β-catenin and glycogen-synthase kinase-3 β (GSK-3 β) signaling were associated accelerated cardiac fibrofatty remodeling in the myocardium of transgenic murine carrying human DSP mutation in response to endurance exercise [100].

In summary, ACM is an inherited disease of heart muscle characterized by life-threatening VTAs and SCD with complex pathogenesis involving progressive fibro-fatty replacement of ventricular myocardium and failing heart. Deleterious mutations in *TMEM43* have been identified to cause severe ACM or ARVC5.

## 4. *TMEM43* Mutations Causal for Arrhythmogenic Cardiomyopathies

Numerous pathogenic variants in different genes with variable penetrance have been identified as a cause for ACM, while the ACM expressivity is highly variable, even among those from the same family or those carrying identical pathogenic variants [72]. An autosomal dominant, lethal and sex influenced ACM with a disease-associated haplotype on chromosome (Chr) 3p was first discovered in a genetically isolated population of Newfoundland and Labrador, Canada [6]. The mutation carried on all recombinant arrhythmogenic haplotypes from these affected families was p.S358L mutation in the *TMEM43* gene. Clinical outcomes of affected individuals have shown that this mutation is fully penetrant. Clinical presentations involved premature ventricular contractions (PVCs), poor R wave progression (PRWP), and LV dilation that results in HF, life-threatening arrythmias, and SCD [6,101]. The mutation affects males more than females, reaching full penetrance earlier and resulting in sudden death for males earlier than females; an extremely high rate of SCD of 48% has been reported in affected male patients. Moreover, the implantable cardioverter defibrillator (ICD) is indicated for primary prophylaxis in postpubertal males, while >/=30 years of age is considered for ICD in females carrying the S358L mutation [102]. This phenomenon was associated with levels of serum testosterone that may worsen ARVC-related pathologies, such as cardiomyocyte apoptosis and lipogenesis [57,58,62,103]. This *TMEM43*-associated arrhythmogenic pathology has been named as ARVC5, and since then, the S358L mutation has been increasingly identified in patients and families with SCD and HF from many other countries, all sharing a common ancestral haplotype [7,104,105,106,107,108]. Most survivors of SCD develop fibrosis with fatty infiltrations in the RV at later stages of life, while some patients develop left-sided or biventricular fibro-fatty pathologies [101,102]. Female *TMEM43* S358L mutation carriers have a more benign phenotype with a later onset and longer survival with the median life expectancy of 71 years of age. Molecular studies have demonstrated that the *TMEM43* S358L causes impaired cardiac function or a fatal arrhythmia as result of increased nuclear stiffness, perturbed *N*-linked glycosylation with accumulation of mutant TMEM43 S358L protein in the nuclear envelope, and decreased conduction velocity at the intercalated discs of cardiac myocytes, which disrupt cell–cell connections, and localization and trafficking of proteins [21,34,92]. Further, cardiac lipid overload via PPARγ signaling and fibrosis through various signaling involving TGF-β3, NF-κB and WNT/β-catenin pathways have been reported [109,110,111].

Fetal ARVC associated with double mutations p.V89M and p.R299T in *TMEM43* displayed RV aneurysm with regionally decreased systolic function and ventricular arrhythmias with frequent PVCs, left bundle branch block, and epsilon waves in the right-sided leads on electrocardiogram (ECG) tracings [112]. Although both variants have been identified in normal controls, double *TMEM43* mutations are extremely rare, acquiring a high-risk for severe ARVC. Findings of CMR imaging in 14 patients with *TMEM43* variants, including eight carrying the S358L mutation, demonstrated LV systolic dysfunction in 57% patients, RV dysfunction in 29% patients, and subepicardial late gadolinium enhancement in 78% patients [113]. Taken together, pathogenic variants in *TMEM43* are associated with biventricular systolic dysfunction and subepicardial late gadolinium enhancement, which suggests a high prevalence of fibro-fatty remodeling.

## 5. Mechanisms of Fibro-Fatty Replacement of the Myocardium in *Tmem43*-Associated Arrhythmogenic Cardiomyopathies

Ventricular arrythmias are thought to cause approximately 75–80% of all SCD cases [114,115], and overweight and obese patients are at increased risks for atrial and ventricular electrophysiological abnormalities, ventricular arrhythmias, HF, and SCD [116]. Although variable amounts of intramyocardial fat may be seen in normal healthy hearts [117], excessive amounts of fat infiltration in the epicardial and intramyocardial tissues have been associated with cardiac arrhythmias [118,119]. Epicardial adipose tissue (EAT) is interposed in the epicardium, which lies between the visceral pericardium and myocardium. The RV has approximately 3–4 times more EAT than the LV [117,120]. The amount of EAT decreases from the base to apex and anterior to posterior surface of the heart. Thickness of EAT correlates with visceral fat mass (not subcutaneous) and is related to body mass index (BMI) and waist circumference [121]. Importantly, the lack of a physical barrier between EAT and the myocardium means EAT can infiltrate into the intramuscular tissue.

Adipogenic conditions are thought to play an important role in the formation of arrhythmia substrates in ARVC5 induced by *TMEM43* [62]. In fact, fatal arrythmias can occur before significant structural abnormalities and fibro-fatty remodeling [62], while build-up of fibro-fatty deposition in the myocardium leads to disarray of the cardiac myocytes, impairing electric stability, and mechanical function of the ventricles [122]. Histologically, abnormal fibro-fatty replacement of the myocardium progresses from the epicardium to the endocardium, particularly affecting the “triangle of dysplasia” in the RV [123]. Fibro-fatty remodeling also includes preferential loss of compact myocardium due to myocyte atrophy and inflammatory infiltrates [124]. *TMEM43* was found as a potential target gene of adipogenic transcription factor PPARγ, which indicates a link between fibro-fatty replacement of cardiomyocytes in the hearts of patients positive for S358L mutation [6,95]. Further, PPARγ-driven fibro-fatty replacement with ventricular cardiomyocyte damage and abnormal electrical activity may be activated by oxidized low-density lipoprotein (oxLDL), which leads to cell internalization of CD36 and upregulation of PPARγ [125,126]. Ultimately, increased lipid accumulation has shown to directly damage cardiac myocytes [125], by perturbing calcium handling [127], sodium currents [86], and contractility [49,82,119,126,128]. ACM patients with high serum levels of oxLDL, including those carrying the *TMEM43* mutation, have worse clinical phenotypes presenting as higher fat displacement of myocytes, ventricular dysfunction, and risk of having a severe, potentially lethal cardiac arrhythmia [113,126].

An important source of fibro-fatty infiltrates in ARVC have been identified as “adipocyte progenitors,” which express the second heart field markers Isl1 and Mef2C [129]. As a candidate “progenitor”, the mesenchymal stem cells (MSCs) have also been studied [62,122]. Co-culture of the *TMEM43*-S358L cardiac myocytes with MSCs was not sufficient to produce significant effect on impulse conduction velocity (CV) or action potential (AP) duration in cardiomyocytes alone. Exposure of co-cultures to pro-adipogenic factors for 2–4 days significantly decreased both CV and AP duration (49% and 31%, respectively) in cardiac myocytes and significantly reduced expression of insulin-like growth factor 1 (IGF-1) in MSCs compared to control non-adipogenic conditions. Importantly, the upregulation of arrhythmogenic substrates occurred long before evidence of structural abnormalities seen in cardiac myocytes, suggesting key roles of adipogenic conditions for developing fibro-fatty infiltrations in ventricular myocardium [62].

## 6. *TMEM43* and Exercise in Arrhythmogenic Cardiomyopathies

Exercise levels are commonly considered in all heart diseases for both the positive and negative effects exercise can have on the type, stage, and progression of the disease [55,130]. Due to the frequent association of symptomatic progression and even SCD due to exercise, much research has been carried out examining how cardiac pathologies and exercise are related [53,54,59]. The relationship between ACM and levels of physical activities is complex, because exercise is typically accepted as a disease accelerator rather than that competitive athletes may have higher risks for developing ACM. Patients with ACM who participate in competitive athletics have been shown to have a five-times greater risk of SCD compared to non-athletes, while the risk of SCD is 5.4 times higher during competitive sports in individuals with ACM than during sedentary activity [84,131]. Current guidelines recommend that ARVC patients participate in neither competitive nor recreational sports due to this tremendously increased risk [49,132]. However, it does not appear that exercise restriction alone is sufficient to prevent unfavorable outcomes, indicating that it is just one of numerous modifying factors in this complex pathogenesis. Additionally, research on the *TMEM43*-genotype ARVC has shown a correlation between increased exercise levels and poorer outcomes for patients, but these studies usually target a younger population based on the assumption that there are higher rates of sports participation and physical activity among this population [107]. Therefore, this is an important factor to consider in evaluating the research focused on ARVC5 and exercise.

The exact mechanism and associations with exercise-induced SCD in ARVC are still unclear in general. On a molecular level, several studies have used exercise to see how overall disease progression differs depending on genetic origin. Study by Martherus et al. examined exercise in transgenic R2834H *Dsp* (Dsp^R2834H^) mice predisposed to ARVC [100]. Exercise induced a faster progression of some of the classic pathological changes seen in the disease, such as RV dilation and wall thinning. Fibro-fatty wall infiltrates and cellular aggregates of cell–cell connecting proteins were also seen. Another line of research from Fabritz et al. proposed that prolonged ventricular activation time in exercise may contribute to the arrhythmia risk [133]. Interestingly, the transgenic *Dsg2* mice in the endurance exercise group in this study again showed a similar cardiac phenotype as in Martherus et al. 2016 study with increased RV size and decreased RV function [100]. However, there was less obvious fibrosis or inflammation. Decreased preload therapy helped prevent the training-induced phenotype in *Dsg2* mice. Mice deficient of JUP also promoted this phenotype, and endurance training accelerated the development of ARVC [134]. These are distinct indications that the increased workload to the heart not only promotes progression of the cardiac ACM phenotype but is also a source of potential prevention and therapy. While there does seem to be a clear throughline that exercise negatively affects cardiac remodeling and promotes ACM, studies disagree on presence and degree of histological changes, which is possibly due to utilization of different animal models for comparison and firm conclusions. Therefore, further research is still needed to elucidate the exact effects and pathways that an exercise and endurance training accelerate and impact the ACM course and cardiac remodeling.

While the molecular mechanism is still unclear, clinical understanding has a stronger foundation with a considerable amount of research concerning ACM and exercise intensity. Several studies have shown that sports triggered life-threatening VTAs during physical exercise among athletes with ACM, predisposing to both sudden death and SCD [135,136]. On the other hand, low-to-moderate intensity exercise presents similar outcomes to that of sedentary patients with ACM, suggesting the benefits of low-to-moderate physical activities in ACM patients without excess risk. Therefore, intense sports are recommended to be avoided in mutation-positive symptomatic ACM patients and in asymptomatic mutation carriers. Although there is a strong link between competitive sports and increased penetrance of ACM-associated mutations [137], more focused research on certain genotypes is critical due to the substantial differences in underlying disease progression and pathology in these patients. It is also important to consider that several of these studies differ in how they examine the actual cardiac outcomes and the population to which they apply. Similarly, the range of patient age at presentation from young adult to middle aged or older necessitates more focused studies on certain demographics [138]. For instance, some of the studies involved a large cohort of healthy athletes [53,139]. While this approach is helpful for establishing risk stratification and preventive screening on a population level, it carries challenges to generalize conclusions to those with the *TMEM43* genotype or similar genotypes, such as desmosomal mutations for example. Others looked at participants of competitive and recreational sports who were carriers for ARVC-associated desmosomal mutations, providing better information for the clinical relevance [140]. Participation in endurance exercise increased ARVC penetrance in patients who were carriers of desmosomal mutation, and genetically susceptible individuals had an increased risk of VTAs along with the onset of HF and SCD [55,134].

One longitudinal cohort study has included patients carrying the *TMEM43* S358L mutation, and exercise appeared to worsen outcomes within this ARVC5 cohort [57]. Interestingly, although there was a higher rate of VTAs in male patients than female patients, this finding did not hold true when adjusting for exercise dose. Generally, males are found to have a stronger association with exercise than females, which is usually attributed to higher rates of athletic participation in males, especially in organized sports at a younger age. However, there has been conflicting information about this assumption, and one study found a higher association with females [52,141]. The study examined 80 patients with the *TMEM43* p.S358L mutation by assessing high level exercise with ≥9.0 metabolic equivalent of task (MET) hours/day and moderate level physical activity with <9.0 MET h/day [107]. The study demonstrated that high level exercise was associated with an increased risk of malignant VTAs, suggesting that this important factor that must be tested to inform exercise effects in ARVC5 patients.

*TMEM43* gene mutations typically cause aggressive disease, requiring VT ablation, and ARVC5 patients have worse composite endpoint of death or heart transplantation [106,142]. Another point to consider is that several studies examined the effects of exercise in ARVC5 patients with an ICD already placed due to progressive disease and severity, and these patients were likely to experience a different effect of exercise on their disease progression [107]. Irrespective of geographical origin, vigorous physical activity has shown to aggravate the ACM phenotype in *TMEM43* S358L carriers with long-term unfavorable prognosis [52], necessitating more *TMEM43* focused research on how exercise affects patients at different stages of the disease.

In summary, deleterious variations in the *TMEM43* gene cause ACM with the S358L mutation causing the most severe arrhythmogenic phenotype. The fully penetrant nature, sex- and exercise-dependent increased risks of clinical outcomes in S358L-mutation positive individuals necessitate personalized risk stratification, comprehensive genetic testing, counseling, and optimized clinical management of affected patients and their families aiming to prevent malignant arrhythmias and SCD. Furthermore, understanding the complex pathogenic mechanisms underlying TMEM43-induced ACM using preclinical cellular and animal modeling and translational research is crucial for developing targeted gene therapies.

## 7. Animal and In Vitro Modeling of the *TMEM43* S358L Mutation

The human TMEM43 protein is 93% identical to the murine protein, making mice ideal models for studying genetic mutations to this protein [1]. Pardon-Barthe et al. completed a murine study that sought to understand ARVD5 disease development and progression, as well as localization, function, and mechanisms of TMEM43 [111]. Transgenic mice overexpressing wild-type (WT) and mutant S358L *TMEM43* in postnatal cardiomyocytes were generated under the control of an alpha-myosin heavy chain (*a*-MHC) promoter. The study found that mice overexpressing the S358L mutation recapitulated human ARVC5 and died at a young age while showing cardiomyocyte death and fibrofatty replacement on a cellular level. It was found that WT TMEM43 was localized to the nuclear membrane and interacted with emerin and β-actin in control mice. On the other hand, mice with the mutation showed partial delocalization of the mutant S358L TMEM43 to the cytoplasm, less interaction with emerin and β-actin, and activation of GSK-3β. Interestingly, overexpression of the calcineurin splice variant, Aβ1, resulted in GSK-3β inhibition, leading to improved cardiac function and survival of mice with the mutation. The study further analyzed the effect of calcineurin Aβ1 in human induced pluripotent stem cells (hiPSCs) bearing the S358L mutation, which showed restoration of contractile dysfunction after GSK-3β inhibition [111]. Moreover, reduced CV and AP duration coincided with a significant reduction in expression of IGF-1 to pro-adipogenic factors, and these arrhythmia substrates were lessened by the IGF-1 treatment [62], suggesting that enhanced adipogenic signaling via IGF-1 signaling may play an important role in the formation of arrhythmia substrates in ARVC. However, the role of IGF-1 underlying early arrhythmogenesis in ACM still needs to be determined.

Shinomiya et al. generated knock-in (KI) mice with the *Tmem43* S358L by using CRISPR-Cas9 targeting and human iPSCs (hiPSC) derived from affected individuals with the S358L mutation [21]. Using both the KI mice and the hiPSCs, the team discovered accumulation of mutant *TMEM43* protein in the nuclear envelope predominately rather than the ER. Under ER-related stress, such as aging and pharmacological stimulation conditions, the membrane topology of some of the *TMEM43* proteins was altered, resulting in glycosylation. Reduction in regional differences in gene expression have been linked to fatal arrhythmias in other mice models of cardiomyopathy [143]. Regional differences in genetic expression between the inner and outer layers of the myocardium in mice were thus studied. While KI mice with the mutation did not show SCD or worse prognosis compared to the WT mice, the hearts of mice with the mutation had reduced regional differences in gene expression [21]. The authors’ proposed the reason may be two-fold: (1) exposing the *TMEM43* KI mice to long-term endurance exercise; and (2) a possible lower probability of ventricular arrythmias occurring at the onset of disease due to small size of mouse hearts, based on previous reports that demonstrated self-termination of electrically induced ventricular arrythmias in rodents [21,144]. Notably, the mice with the mutation in this study had marked age-dependent decline of the ER function and fibrotic replacement of the subepicardial myocardium.

Another group by Stroud et al. has created a similar *Tmem43* S358L KI model and germline null *Tmem43*^−/−^ or *Luma*^−/−^ mice using the same CRISPR-Cas9 strategy [145]. The authors describe utilizing embryonic TMEM43-mutated stem cells which were implanted in female mice and chimeric mice were then crossed with Sox2-Cre mice to obtain *Luma*^+/−^ mice. Both mutant S358L and *Luma*^−/−^ mice displayed normal cardiac function and morphology and responded normally to pressure overload induced by transverse aortic constriction. Moreover, LINC complex proteins and their localization and expression in cardiac myocytes and fibroblasts were unaffected by global loss of *Luma* or S358L mutation. The authors suggested that the S358L mutation alone may not cause cardiomyopathy in vivo.

Using CRISPR/Cas9 approach, Klinke et al. illustrated the importance of this highly conserved protein in a study using *Drosophila melanogaster* [20]. In this study, a knock-out (KO) line of *Drosophila* was generated in addition to a mutant line that overexpressed CG8111-S333L variant, which was homologous to the human *TMEM43* S358L mutation. The KO flies developed normally, while those with overexpression of the CG8111-S333L mutation had growth defects, cardiac arrhythmias, loss of body weight, and premature death. The study found that conservation of serine was critical for physiological function of the gene by replacing S333 with select amino acids. Further analyses revealed that S333 in CG8111 was essential for lipid metabolism and energy homeostasis.

Zheng G found that the *Tmem43* S358L mutation leads to increased activation of NF-kB and TGFb signaling in heart tissues of mutant mice, which is generated using conventional gene targeting of TCC->TTA substitution in C57BL/6 mouse [110]. These S358L KI mice developed a higher level of both fibrosis and adipocyte formation in the hearts and an increased expression of fibrosis and adipogenesis markers were detected, while inflammatory responses were minimal. The authors explained the lack of inflammation as a result of anti-inflammatory effects of up-regulated PPARγ pathway in mutant mice [146].

Gu et al. generated a *TMEM43* KI model by targeting exon 12 in the *Tmem43* gene and generated the p.S358L mutation by changing two bases at c.1072T->C and c.1073C->T by using a homologous recombination method [147]. Chimeric animals generated were crossed with C57BL/6J mice, and mice with the mutation were utilized for systems genetics studies by comparing with a large family of murine genetic reference population (GRP) of recombinant inbred (RI) BXD mice derived from two parental strains, C57BL/6J and DBA2/J [148]. This study investigated systems biological roles of *TMEM43* through genetic regulation, gene pathways and gene networks, candidate interacting genes, and up-stream or downstream regulators. They found high *Tmem43* levels with broad variability in expression in the heart tissue among the BXD mice. Expression of cardiac *TMEM43* was negatively correlated with heart mass and heart rate, while levels of *TMEM43* were positively associated with levels of plasma high-density lipoproteins (HDL) among BXD mice and defined importance of *TMEM43* for cardiac- and metabolism-related pathways. Similarly, we utilized these mice and demonstrated that mice with both heterozygous and homozygous S358L mutations displayed arrhythmias, fibro-fatty infiltration, and mitochondrial abnormalities in the myocardium [109].

Our group also demonstrated that the S358L mutation affected not only cardiac functions, but also small intestine functions in KI mice with the mutation and metabolic homeostasis via abnormal WNT/β-catenin and PPARγ signaling, which suggested the importance of TMEM43 protein regulation in the small intestine and metabolic disturbances to ARVC5 pathophysiology as results of dual heart–gut functions of TMEM43 as shown in Figure 6. In brief, the study demonstrated a significant increase in lipids in the jejunum of mice with the *Tmem43* S358L mutation, which was like fibro-fatty infiltrations seen in the myocardium of the mice. We also found that feces from mice with the mutation contained 35% less lipid compared with WT littermate mice, while plasma LDL levels were significantly higher in animals with the mutation, suggesting that these mice absorbed lipids and fatty acids more efficiently from the gut lumen. Further, expression of Ki-67, β-catenin, and the genes downstream to PPARγ was significantly increased in the gut mucosa of mice with the *Tmem43* mutation, suggesting that the S358L augmented gut epithelial proliferation, increasing macronutrient and fat uptake surface and capacity. These results provided not only a mechanism of systemic hyperlipidemia in mice with the mutation but also suggested an association between TMEM43 S358L and fibro-fatty remodeling of the myocardium and ACM progression (Figure 6). In support of this phenomena, increased fat absorption from the gut lumen in obesity, diabetes, and metabolic dysfunction-associated steatotic liver disease (MASLD) have a strong association with VTAs [149,150,151]. Moreover, as noted earlier ACM patients with high serum levels of oxLDL have worse clinical phenotypes with higher fatty remodeling in the myocardium, ventricular dysfunction, and risk of having a severe, potentially, lethal cardiac arrhythmia [125,126].

An alternative animal model to rodents was transgenic zebrafish developed by Zink et al. using the Tol2-system [37]. The group developed a zebrafish model that overexpressed eGFP linked with full length human *TMEM43* WT and two mutations of either S358L or p.P111L. The P111L mutation and overexpression of TMEM43 WT protein were associated with transcriptional activation of the mechanistic target of rapamycin (mTOR) pathway and ribosome biogenesis. These signaling modifications led to an enlarged heart and cardiomyocyte hypertrophy, a possible precursor to the ARVC pathway. However, structural function was not altered in these two lines. On the other hand, the S358L mutation resulted in an instability of the nuclear membrane TMEM43 protein, which was partially redistributed into the cytoplasm in embryos and adult zebrafish. The zebrafish with the S358L mutation showed age-dependent heart enlargement and structural remodeling and transcriptional alterations, which may play a role in the pathogenesis of these changes. No detection of fibro-fatty replacement or abnormal electrical activity was found in the hearts of zebrafish with either mutation, which may be due to the lack of cardiac mechanical stress in zebrafish models. Another possible reason is simply differences in TMEM43 expression between species. Nonetheless, the conserved nature of the TMEM43 protein combined with its consistent cardiac effects underscores the importance of continued animal model research to discern the mechanisms of ARVC caused by mutations in the *TMEM43* gene.

Other TMEM43-related cardiac pathologies include reduced expression of *Tmem43* during cardiac hypertrophy that leads to worsening heart failure in mice [16], while haplo-deficiency of *Tmem43* that results in age-dependent late-onset senescence-associated cardiomyopathy via the DDR pathways [15]. Normal cellular morphology was demonstrated in patient’s iPSC line, which carried a heterozygous intronic splice *TMEM43* variant—c.443-2A>G [152]. As a transmembrane protein, TMEM43 has been studied as possibly being involved in ferroptosis and its role in LPS-induced cardiac injury [17]. The study demonstrated that *TMEM43* protects against sepsis-induced cardiac damage by inhibiting ferroptosis in mice, which suggests the use of *TMEM43* as a therapeutic target for preventing sepsis.

Related to therapeutic targeting, the most recent findings by Lalaguna et al. show that overexpression of the *Tmem43* WT gene in both WT mice and mice with the S358L mutation caused improved cardiac function, which was especially prominent in the cardiomyocytes, with the use of an α-MHC promoter [153]. In this study, the authors first created double transgenic mice by crossing mice with overexpression of TMEM43 WT and TMEM43 S358L. Through a 24-week analysis of echocardiography and ECG, they found that the overexpression of the *Tmem43* WT gene could combat the effects of the S358L mutation as these double transgenic mice were able to live around 10 weeks longer compared to mice with the S358L mutation. Secondly, the authors described administering an adeno-associated virus (AAV) carrying *Tmem43* WT in mice with the S358L mutation and showed that cardiac function was restored in a dose-dependent pattern, suggesting that gene therapy via AAV-mediated delivery could offer beneficial effects by delaying ARVC5 development and mitigating the disease progression in vivo. The study demonstrated that gene therapies by overexpressing or delivering the *TMEM43* WT gene with an AAV-mediated strategy to replace or dominate over the *TMEM43* mutation function, offer an effective and personalized treatment that targets the underlying primary cause of ARVC5 [153,154].

In summary, the S358L TMEM43 mutation causes severe cardiac arrhythmias, SCD, and progressive HF in animal models characteristic of human ARVC5 by disrupting the structural integrity, cell-to-cell adhesion, and gap junction function of cardiomyocytes. Moreover, the S358L TMEM43 mutation is shown to increase nuclear stiffness and promote cell death. Critical signaling pathways have shown that disrupted WNT/β-catenin and reduced PPARγ activity underlie fibro-fatty remodeling of the myocardium, fostering re-entrant VTAs. These mutation-induced alterations extend to small intestine, resulting in elevated lipid absorption and hyperlipidemia contributing to fibro-fatty build-up in the heart and small intestine. Gene-targeting therapeutic strategies by overexpressing TMEM43 WT or by replacing mutant TMEM43 with WT protein have demonstrated remarkable promise in preclinical animal studies.

## 8. TMEM43 and Muscular Dystrophy

Skeletal muscle system is one of multiple organs affected by TMEM43. Mutations in *TMEM43* resulting in emerin degradation or its improper delocalization from the nuclear membrane resulting in disruption of the LINC complex have been implicated in the development of EDMD-related myopathy, which may present clinically with muscular dystrophy, joint contractures, and cardiomyopathy [1,8]. Two heterozygous missense mutations in *TMEM43*, p.E85K and p.I91V, have been identified in two patients with EDMD-related myopathy. In the patient with the p.E85K mutation, reduced nuclear staining of TMEM43 was observed in myocytes. This mutation also resulted in failure of *TMEM43* to self-oligomerize, which is a process postulated to be important for protein complex formation [8]. An EDMD-related myopathy sign due to the other mutation was early toe-walking—known as Pes Equinus—in a father and his son who were carriers of a heterozygous E85K variant. On physical exam, both patients had multiple joint contractures, axial and proximal muscle atrophy, and proximal weakness. Neither the father nor son had a high arched palate or facial involvement. Both patients also had diffuse muscle atrophy and fatty infiltration as shown on CT, although in different muscles. Both patients had an elevated creatine kinase. The father had a muscle biopsy of the left biceps brachii, which showed fatty infiltration, mild necrosis, and regenerative fibers. Hematoxylin and eosin stain revealed moderate variation in the size of muscle fibers. Immunohistochemical strains for dystrophin, caveolin, dysferlin, alpha-dystroglycan, beta and gamma sarcoglycan, collagen VI, emerin, and laminin a-2 chain showed no abnormalities [9]. For cardiac involvement, both patients had normal echocardiogram, Holter monitor, and ECG results, although the son was diagnosed with atrial fibrillation at 28 years old. It was proposed that the development of ARVC5 in EDMD-related myopathy is patient-dependent, since neither patient developed cardiomyopathy. Another gene of the LINC complex that causes an EDMD spectrum disease is *SYNE2*/nesprin 2 that is involved in regulation of the structure and function of the nuclear envelope [155]. As TMEM43 is responsible for causing EDMD-7, mutations in *SYNE2* cause EDMD-5 [156]. This study demonstrated that the pathogenic mechanism of a missense *SYNE2* p.S4126R mutation-induced EDMD-5 condition involves disruption of the LINC complex interactions with nuclear envelope proteins including emerin and SUN2, ultimately leading to reduced expression of the nuclear membrane proteins and their delocalization from abnormally shaped nuclei of skeletal muscle cells, causing EDMD-related myopathy phenotypes.

## 9. TMEM43 and Cancer Progression

The TMEM family includes proteins of mostly unknown functions, while many TMEMs have been implicated in several different cancers as tumor suppressors or oncogenes [157]. While the presence or overexpression of TMEMs is generally considered a risk factor for many cancers, genetic analysis has shown that the TMEM88 protein has diverse functions in cancers, such as protective effects in gliomas [158], while it promotes invasion and metastasis in lung cancers and predicts the survival and palindromia time of hepatocellular carcinoma (HCC) [159,160,161]. In lung cancer, TMEM48 and TMEM97 were described as potential prognostic biomarkers [162,163], *TMEM25* gene expression was associated with colorectal cancer [164], accumulation of TMEM176 protein was associated with increased in lymphomas [165], and TMEM7, which suppresses cell proliferation, was downregulated in HCC cells [166].

Looking specifically at the mechanisms behind TMEM43’s role in cancer development reveals several different potential areas of further study [12]. In small cell lung cancer (SCLC), a highly aggressive disease, univariate COX regression analysis and random survival forest analysis examining a miRNA-mRNA network has shown that TMEM43 decreases overall survival [167]. *TMEM43* was identified as one of the network genes associated with the decreased overall survival of SCLC and a possible explanation for this was TMEM43’s involvement in LINC complex function, which is critical to cell migration in highly invasive SCLC. Another study using linkage analysis and whole exome sequencing performed in 39 patients with the Serrated Polyposis Syndrome (SPS) identified *TMEM43* as a possible germline SPS predisposition gene [168].

A more detailed mechanistic understanding of several types of cancers has provided several routes for possible therapeutic and prognostic exploration. Li et al. investigated *TMEM43* expression levels in human pancreatic cancer versus control samples [14]. Immunoassays, co-immunoprecipitation, and protein mass spectrometry were utilized for further analysis. The authors found elevated *TMEM43* protein expression levels as well as a correlation with poor disease-free and overall survival in patients. Knock-down of *TMEM43* in vitro inhibited pancreatic cancer progression, dysregulated the cell cycle, and promoted tumorigenicity in vivo. PPRF3 protein levels may be one possible reason for this, but mRNA levels did not differ between *TMEM43* knockdowns and controls. The RAP2B/ERK axis transcription may be the downstream effect of *TMEM43* in pancreatic cancer.

Tumor development was also affected by *TMEM43* expression levels in HCC. RNA sequencing and The Cancer Genome Atlas (TGCA) databases were used to explore genes in HCC, and TMEM43’s role was explored through cell counting kit 8 (CCK8) cloning, flow cytometry, and Transwell experiments [13]. *TMEM43* was shown to be highly expressed in HCC and the absence of TMEM43 in cancer cells has been reported to have a negative association with HCC progression via ubiquitin-specific protease 7 (USP7) and voltage-dependent anion channel 1 (VDAC1), a protein that regulates entry of molecules into the outer mitochondrial membrane. The interplay between TMEM43 and VDAC1 was also explored as a possible regulatory interaction, and TMEM43 appeared to activate VDAC1 through USP7 de-ubiquitination, representing a new predictive and treatment target for HCC.

TMEM43 was shown to be a key component of the EGFR pathway to recruit TMEM43 following epidermal growth factor (EGF) stimulation [12]. TMEM43 then interacts with CARMA3, a scaffold protein, leading to downstream NF-kB activation that plays a key role in controlling cell survival. The study also showed TMEM43 affected migration and invasion of cancer cells in vitro and tumor progression in vivo. Higher *TMEM43* gene expression was also found to correlate closely with brain tumor malignancy, while suppression inhibits tumor growth.

Taken together, the pathogenesis of these different cancers illustrates the varied mechanistic processes of which TMEM43 can alter cancer pathogenesis. TMEM43’s role in pathogenicity in cancer is vital, but further research is still needed to provide clinical translation. Lei et al. discussed three challenges that should guide further research into cancer progression through TMEM effects: (1) the roles of TMEMs and their varied roles are unclear; (2) the 3D structure, epitopes, active sites, and affinities of TMEMs have not been fully described; (3) TMEMs are difficult to express/extract due to their lipophilic transmembrane properties [169]. These three areas should serve as critical areas of exploration in the future.

## 10. TMEM43 and Auditory Neuropathy Spectrum Disorders

Hearing loss is one of the most common and devastating auditory neuropathy and sensory disorders [170]. Hearing impairments and loss in ANSDs are characterized by deteriorated speech perception and abnormal auditory brainstem responses in affected patients, while otoacoustic emissions and pure-tone detection are normal. Therefore, mechanisms of ANSD-related hearing loss include electrophysiological dysfunctions at the cochlear inner hair cells and spiral ganglion neurons, while the function of outer hair cells is preserved. In the organ of Corti of the inner ear, TMEM43 is shown mainly to be expressed in the plasma membrane of cochlear GLSs [5,10,171], which are a major supporting cell type of auditory system that reside adjacent to hair cells of inner ear and express glia markers, such as glial fibrillary acid protein (GFAP) and the glutamate-aspartate transporter (GLAST) [172]. GLSs play crucial roles in the development, maintenance and function of auditory system. These cells also have an impact of hair cell regeneration, causing disorders of inner ear and ANSDs.

Jang et al. identified the nonsense p.R372X mutation that leads to a truncated TMEM43 protein lacking the 4th TM domain in two Asian families with histories of late-onset ANSD and progressive hearing loss. Members of these multigenerational families manifested no symptoms of either ACM or any other cardiac abnormalities [10]. A mouse KI model expressing the truncated TMEM43 protein lacking the last 29 amino acids displayed ANSD phenotypes with progressive hearing loss. The molecular mechanisms underlying of ANSD has been explained by the fact that TMEM43 interacts with the CX channels via CX26 and CX30 in gap junctions of cochlear GLSs and the mutation disrupts passive conductance current in GLSs. Further, the same research group demonstrated that TMEM43 interacts with two pore domain potassium (K2P) channels by forming the protein-protein complex with TASK-1, encoded by the KCNK3 gene [5,173]. Perturbance of both TMEM43 and TASK-1 has shown to cause dysregulation of the cochlear K^+^ homeostasis via leak K^+^ channels and gap junction channels, leading to sensory dysfunction, auditory neuropathy, and hearing loss [10,174]. To support human and mouse findings, hiPSC lines (UMi040-A) derived from a patient lymphoblastoid cell line carrying a single TMEM43 Arg372Ter mutation have been developed to study R372X-induced disease pathogenesis [11]. In addition, a TMEM43 KO hiPSC line (HDZi003-A-1) has recently been generated using the CRISPR/Cas9 genome editing [175]. These hiPSC lines demonstrated a deficiency of TMEM43 and showed normal morphology and a stable karyotype. The resulting cell lines had a deficiency of TMEM43 and showed normal morphology and a stable karyotype. Recently, an important role for TMEM43 in gap junction networks in the brain has been reported [171]. TMEM43 KO mice displayed increased neuronal excitability in the hippocampus associated with decreased K^+^ uptake in the gap junction-coupled syncytium of astrocytes. All these discoveries suggest that investigation whether TMEM43 is involved in forming channels in cardiac and skeletal myocytes and whether mutation in TMEM43 directly affects function or distribution of various ion channels may open new avenues for developing targeted therapies for many TMEM43 mutation-induced devastating disorders, including ARVC and EDMD-related myopathies.

## 11. Conclusions

TMEM43 is a key protein in the nuclear membrane structure that serves as a key connector of the cellular nucleoskeleton and cytoskeleton. TMEM43 is differentially expressed in different tissues, including the intercalated discs in cardiac myocytes, a likely contributor to the pathogenesis of various diseases such as arrhythmogenic cardiomyopathy, skeletal myopathies, cancers, and auditory neuropathy spectrum disorders. There are multiple mechanistic theories for the *TMEM43* S358L mutation in ARVC5 with severe ventricular tachyarrhythmias, exercise-related risks of sudden death, and unfavorable long-term prognosis, including the effects of pro-adipogenic factors and the heart–gut axis in the development of fibrofatty infiltrations in the myocardium. Multiple animal models and in vitro strategies have been employed with differing degrees of success. However, the molecular mechanisms remain unclear, requiring further in-depth TMEM43-focused research for understanding its role in the clinical relationship between ARVC5 and exercise and to fully unravel drug and gene targeting therapies for TMEM43-related diseases.

## Figures and Tables

**Figure 1 ijms-26-06856-f001:**
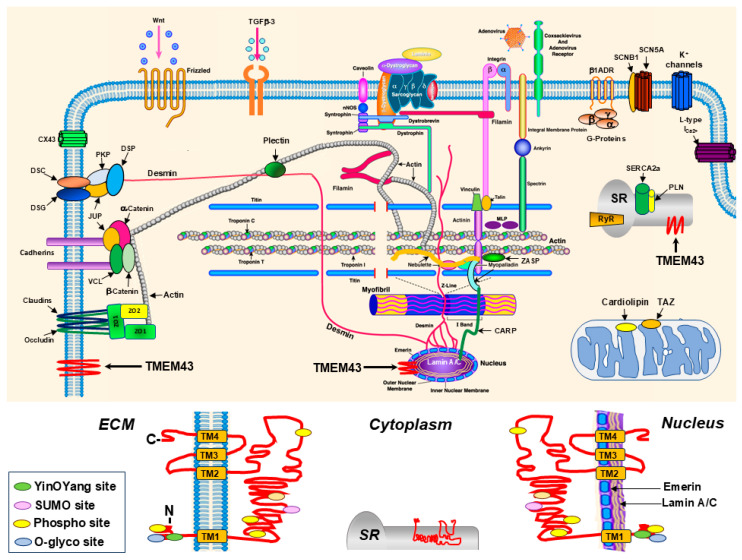
Schema of the cardiomyocyte structure and TMEM43 protein. Upper panel: Schema of the proteins expressed in different subcellular components of cardiomyocyte. The TMEM43 protein depicted by red zigzag line is expressed in the sarcolemma, inner nuclear membrane, and sarcoplasmic reticulum (SR) of cardiomyocytes. The schema was adapted and improved from Towbin JA. Inherited cardiomyopathies. *Circ J.* **2014**, *78*, 2347–2356 [18]. The genes and proteins identified are updated according to the recent discoveries. Lower panel: Detailed schematic picture of the TMEM43 protein structure and localization of different functional sites across the cardiomyocyte sarcoplasm. The protein’s N- and C-termini of the sarcolemmal TMEM43 are localized in the extracellular matrix (ECM), of the inner nuclear membrane TMEM43 are in the nucleus, and of the sarcoplasmic reticulum (SR) membrane TMEM43 are in the cytoplasmic compartment of the cardiomyocyte. Four transmembrane (TM1-TM4) domains, five phosphorylation (Phospho, yellow) cites, one small ubiquitin-like modifier (SUMO, pink), one YinOYang (green), and one O-glycosylation (O-glyco, blue) cites are identified.

**Figure 2 ijms-26-06856-f002:**
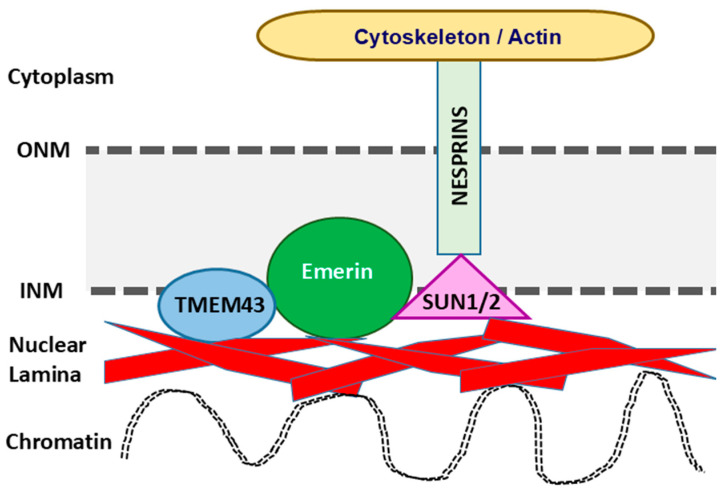
Schema of the nuclear envelope and interactions of TMEM43. TMEM43 interacts with nuclear lamins (red) and emerin (green) in the inner nuclear membrane that are connected to the SUN1/2 (pink) and nesprins (SYNE 1, SYNE2, light green) forming the linker of nucleoskeleton and cytoskeleton complex (LINC). The LINC is connected to cytoskeleton and actin in the cytoplasm. INM, inner nuclear membrane; ONM, outer nuclear membrane.

**Figure 3 ijms-26-06856-f003:**
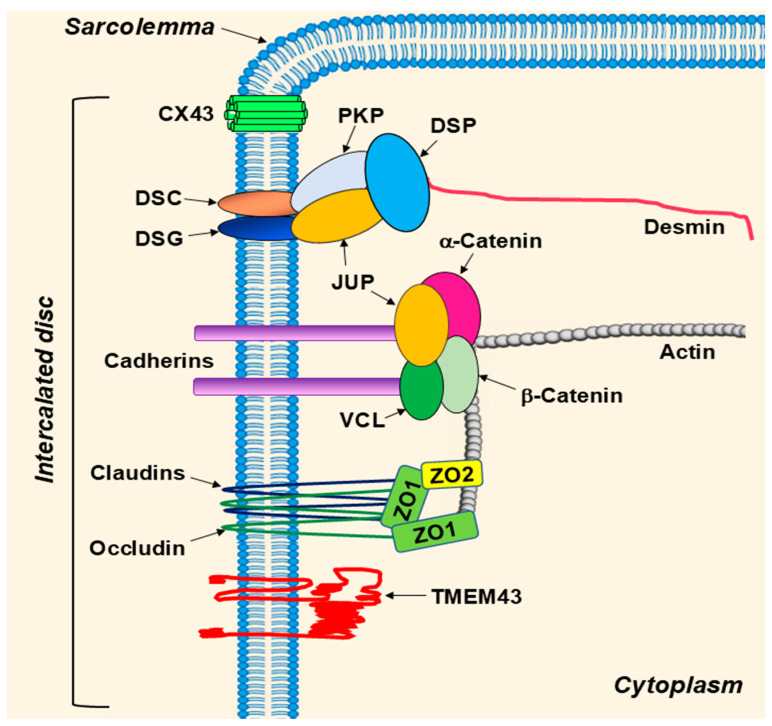
Schema of the cardiomyocyte intercalated disc. Three structures are distinguished in the intercalated disc of cardiac myocytes. Adherens junctions contain cadherins, catenins and vinculin (VCL). Desmosomes are comprised by desmoplakin (DSP), desmocollin (DSC), desmoglein (DSG), plakoglobin (JUP), plakophilins (PKP) and desmin (intermediate filament). Gap junctions contain connexins (CX43 and other CXs). Claudins, occludins, zonula occludens (ZO) and TMEM43 proteins are also identified in the intercalated discs of cardiomyocytes.

**Figure 4 ijms-26-06856-f004:**
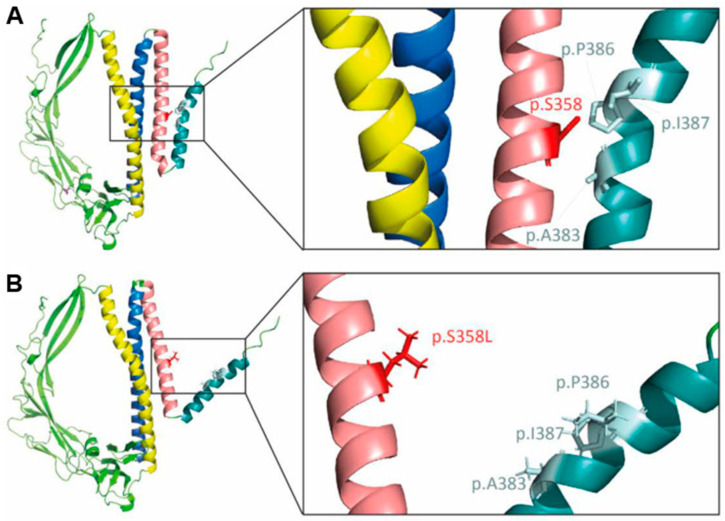
The assembly of 3-D structure of TMEM43 protein using AlphaFold2. (**A**) The tertiary structure of wild-type TMEM43. The S358 residue forms hydrogen bonds between TM3 and TM4 helices per in silico modeling, maintaining proper TMEM43 protein structure. Protein’s transmembrane (TM) domains are color coded: yellow—TM1, blue—TM2, rose—TM3, and turquoise—TM4. (**B**) The prediction shows that the p.S358L substitution disrupts transmembrane helix stability by breaking the hydrogen bonds between TM3 and TM4. Adapted from Zink et al. Altered Expression of TMEM43 Causes Abnormal Cardiac Structure and Function in Zebrafish. *Int. J. Mol. Sci.* **2022**, *23*, 9530 [37].

**Figure 5 ijms-26-06856-f005:**
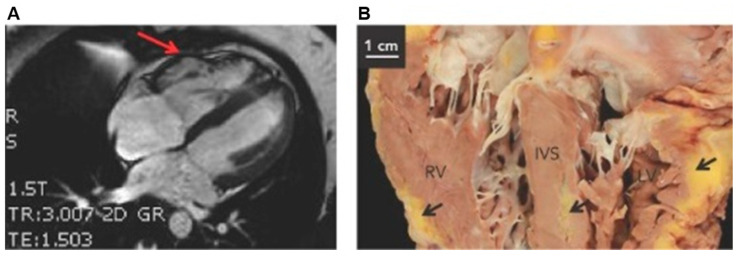
Cardiac pathology shown in a patient with arrhythmogenic cardiomyopathy. (**A**) Cardiac magnetic resonance image demonstrating myocardial wall thinning in the right ventricle (RV, red arrow). (**B**) Fibro-fatty infiltrations indicated by black arrows in the RV, interventricular septum (IVS), and left ventricle (LV). Adapted from *Arrhythm. Electrophysiol. Rev.* **2016**, *5*, 90–101 [89].

**Figure 6 ijms-26-06856-f006:**
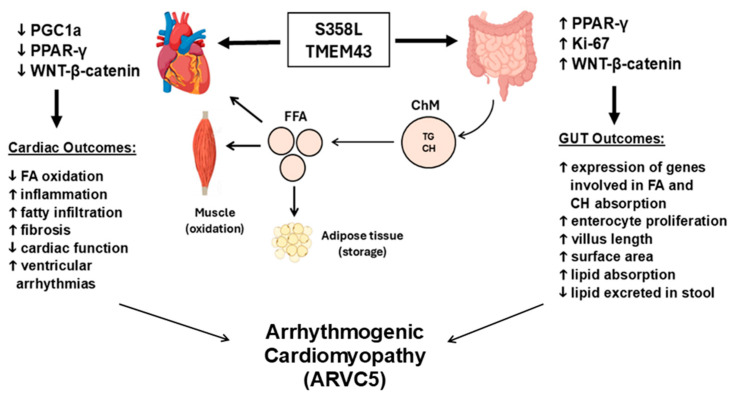
Proposed schema of the heart–gut crosstalk in arrhythmogenic cardiomyopathy caused by the *TMEM43* S358L mutation. Dual opposite effects on heart (downregulation of PPARγ and WNT/β-catenin) and small intestine (upregulation of PPARγ and WNT/β-catenin) pathways were identified in mutant S358L knock-in mice, leading to ARVC5. FA, fatty acids; FFA, free fatty acids; TG, triglycerides; CH, cholesterol; ChM chylomicron; TG, triglyceride. Arrows indicate downstream effects or outcomes.

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
