# Peer review of "Transmembrane Protein 43: Molecular and Pathogenetic Implications in Arrhythmogenic Cardiomyopathy and Various Other Diseases"

_ijms, 2025, doi:10.3390/ijms26146856_

Round 1
Reviewer 1 Report
Comments and Suggestions for Authors
Please, see attached

Author Response
1) Authors provide comprehensive overview of studies on TMEM43. However, as functions of TMEM43 are not completely resolved, in my opinion, the many of manuscript sections would benefit of closing paragraphs that clearly summarize what is already established, what is “proposed” and what are the main future directions.
Response: Thank the reviewer. We have provided a summary at the end of each section.
2) Fig.1 ‘upper panel’ should be in bold and this panel would benefit from more detailed description.
Response: The Figure 1 is modified according to the comment.
3) There are a lot of abbreviations used. I find “Abbreviation” section very useful. However, it is not complete with some abbreviations missing (EAT, EDMD, ANSD and etc.). Also, it would be better if it is arranged in an alphabetical order.
Response: The Abbreviation section is improved.
4) Just out of curiosity: is there any indications that TMEM43 form channels in cardiac myocytes? or if it directly affects function/distribution of other ion channels?
Response: Thanks for the intriguing idea. We added some discussions on this topic, please see Section 10, lines 831-836.
Reviewer 2 Report
Comments and Suggestions for Authors
A thorough and well-written review on TMEM42 on heart and other tissues is presented. Comments are minor in nature:
- Since MEM43 is also highly expressed in placenta, what are the effects of mutations on this tissue and role in placental disorders?
- Most figures can benefit with a brief description, increasing FONT size in some, and in those with several abbreviations, by defining them.
- Comment on the physiological role of TMEM43 in sarcolemmal membranes.
- Figure 3 is somewhat blurry, based on the downloaded version. This can be improved.
- Mutations appear to affect males more than females (page 8, line 286). Assuming this is hormonally based, how does testosterone and/or estrogen contribute (mechanisms) to these differences?
- Page 10. Can the authors be more specific about the type and intensity of exercise mentioned in lines 383+ and provide the link between exercise intensity and VTAs on page 11.
Author Response
A thorough and well-written review on TMEM42 on heart and other tissues is presented. Comments are minor in nature:
1) Since MEM43 is also highly expressed in placenta, what are the effects of mutations on this tissue and role in placental disorders?
Response: Thanks for the positive assessment and for the information that we unnoticed in the original paper. We added literature and some discussions on this topic. Please see Section 2, lines 206-213.
2) Most figures can benefit with a brief description, increasing FONT size in some, and in those with several abbreviations, by defining them.
Response: Figures are improved according to the comment.
3) Comment on the physiological role of TMEM43 in sarcolemmal membranes.
Response: Thank the reviewer. We added information on the physiological role of TMEM43 in sarcolemmal membranes (lines 159-165). The additional Figure 3 is also provided, depicting the intercalated disc structure in the cardiomyocyte sarcolemma.
4) Figure 3 is somewhat blurry, based on the downloaded version. This can be improved.
Response: The Figure 4 in the manuscript is replaced with the image with better resolution.
5) Mutations appear to affect males more than females (page 8, line 286). Assuming this is hormonally based, how does testosterone and/or estrogen contribute (mechanisms) to these differences?
Response: Thanks for pointing this important feature of ACMs in general that we overlooked. We added information on sex-dependent clinical features and outcomes. In addition, research studying mechanisms of testosterone and/or estrogen underlying sex differences in ACM phenotypes is added to the Section 3: Arrhythmogenic Cardiomyopathies (lines 243-252).
6) Page 10. Can the authors be more specific about the type and intensity of exercise mentioned in lines 383+ and provide the link between exercise intensity and VTAs on page 11.
Response: We modified the Section 6 by providing more specific information on associations between exercise intensity and VTAs, please refer to Section 6, lines 439-448 and 481-488.
Reviewer 3 Report
Comments and Suggestions for Authors
The review by Orgil et al. focuses on structure and function of the Transmembrane Protein 43 (TMEM43), and the pathogenetic pathways involved in arrhythmogenic cardiomyopaty and other associated disorders. The topic is "hot" and relevant to the field.
The review is very complete and detailed, however in my opinion it is too long and descriptive. An effort to summarize in a more concise and effective way the published studies should be made. Indeed, the detailed report of each study risks to be dispersive and merely descriptive, eventually lacking synthesis. Rather, a little summary at the end of each paragraph could help to identify the main point, synthesize the state of the art and identify current limitations and future directions in the Authors' view.
Additional points:
- Figure 1: it is not clear what the long red lines connecting nuclear membrane with desmosomes and cytoskeleton represent. A more detailed legend would be useful.
- Figure 3: please describe in the legend what panels A and B represent.
Minor points:
- Line 71: conserved not conversed
- Please rephrase lines 177-182 which are a bit confusing.
Comments on the Quality of English LanguageEnglish language is generally correct but in some parts (ex. lines 177-182) not very fluent. A language revision would be suggested.
Author Response
The topic is "hot" and relevant to the field. The review is very complete and detailed, however in my opinion it is too long and descriptive. An effort to summarize in a more concise and effective way the published studies should be made. Indeed, the detailed report of each study risks to be dispersive and merely descriptive, eventually lacking synthesis. Rather, a little summary at the end of each paragraph could help to identify the main point, synthesize the state of the art and identify current limitations and future directions in the Authors' view.
Response: Thank the reviewer. We have provided a summary at the end of each section.
Additional points:
1) Figure 1: it is not clear what the long red lines connecting nuclear membrane with desmosomes and cytoskeleton represent. A more detailed legend would be useful.
Response: The Figure 1 is modified and improved according to the comment.
2) Figure 3: please describe in the legend what panels A and B represent.
Response: The Figure 3 is improved according to the comment
Minor points:
- Line 71: conserved not conversed: Corrected
- Please rephrase lines 177-182 which are a bit confusing: Corrected.
Round 2
Reviewer 3 Report
Comments and Suggestions for Authors
Overall, I appreciate the effort of the authors to clarify some points in the text, rendering it more effective, and provide more detailed figure legends.
Fig. 3: Although it is a good idea to insert a figure illustrating more in detail the different molecules involved in intercalated discs, this picture is misleading, since it seems that desmosomal proteins participate to adherens junctions by directly interacting with cadherins, whereas desmoglein and desmocollin are transmembrane proteins involved in desmosome formation. A clearer separation between the different types of junctions, as also detailed in the legend, is necessary. The same applies to left part of Fig. 1.
- Line 153: In contraST not in contract
- Line 203: This study not only defined a potential link between TMEM43 expression and…..?
- Line 224: "A diagnosis of ARVC generally occurred after the individual 224 presents with arrhythmia, particularly with SCD in young people [45]. A classical feature of ACMs is the presence of symptomatic arrhythmia and diagnosis generally occurs after the individual presents with arrhythmia" Redundant!
- Line 713: that IS involved in regulation
- Gene names should be in italics
Author Response
Reviewer 3
- Fig. 3: Although it is a good idea to insert a figure illustrating more in detail the different molecules involved in intercalated discs, this picture is misleading, since it seems that desmosomal proteins participate to adherens junctions by directly interacting with cadherins, whereas desmoglein and desmocollin are transmembrane proteins involved in desmosome formation. A clearer separation between the different types of junctions, as also detailed in the legend, is necessary. The same applies to left part of Fig. 1.
Response: We absolutely agree with the comment. Thank the reviewer for the careful review. Figures 1 and 3 have been corrected accordingly.
- Line 153: In contraST not in contract - Corrected
- Line 203: This study not only defined a potential link between TMEM43 expression and…..? - Corrected.
- Line 224: "A diagnosis of ARVC generally occurred after the individual 224 presents with arrhythmia, particularly with SCD in young people [45]. A classical feature of ACMs is the presence of symptomatic arrhythmia and diagnosis generally occurs after the individual presents with arrhythmia" Redundant! - Corrected.
- Line 713: that IS involved in regulation – Corrected.
- Gene names should be in italics – Corrected where this applies.